# Bacterial expression, purification and folding of exceptionally hydrophobic and essential protein: Surfactant Protein-B (SP-B)

Tadiwos Asrat[1], Donna Jackman[1], Valerie Booth[1,2]*

1 Department of Biochemistry, Memorial University of Newfoundland, St. John's, Newfoundland and Labrador, Canada, 2 Department Physics and Physical Oceanography, Memorial University of Newfoundland, St. John's, Newfoundland and Labrador, Canada

* vbooth@mun.ca

## Abstract

Lung Surfactant Protein B (SP-B) is essential for life. It is thus striking that, to this point, no method for making the full-length protein has been published and consequently we lack detailed understanding of SP-B's basic structure-function relationships, as well as an inability to make it for clinical use. The major challenge in producing SP-B lies with its exceptionally hydrophobic nature. In this work, we present a method to produce recombinant SP-B in bacteria that can be used to make the full-length protein as well as the product focused on here, which is a construct lacking the N-terminal 7 residues, rSP-B ($\Delta7_{NT}$C48S-SP-B-6His). The construct is produced as a fusion to *Staphylococcus* nuclease A (SN) in *Escherichia coli* C43 cells, a strain known to promote production of toxic and membrane recombinant proteins. After cleavage from SN, rSP-B is folded on column and then exchanged into the lipid or detergent system of choice. rSP-B prepared in this way exhibits the correct secondary structure and demonstrates surface activity. The yield obtained is 0.3 mg of purified rSP-B ($\Delta7_{NT}$C48S-SP-B-6His) per liter of initial bacterial culture. We expect this method for producing SP-B will be valuable in enabling basic research into SP-B's mechanisms, as well as possibly facilitating the inclusion of SP-B in lung surfactant formulations to treat common and frequently fatal lung conditions and in lung surfactant-based drug delivery.

## 1. Introduction

Surfactant protein B (SP-B) is essential for life via its role in breathing. In the lungs, gas exchange between inhaled air and the blood occurs in alveoli. The alveolar air/water interface is lined with a complex mixture of lipids and proteins (including SP-B) called lung surfactant (LS). LS reduces the high surface tension at the air-water interface, preventing alveolar collapse and reducing the work of breathing [1]. Lack of functional LS in the alveolar space can thus result in lung collapse. For example,

**Data availability statement:** All relevant data are within the manuscript and its Supporting Information files.

**Funding:** This work was funded by a National Sciences and Engineering Research Council of Canada Discovery Grant to VB (204115). The funders had no role in study design, data collection and analysis, decision to publish, or preparation of the manuscript.

**Competing interests:** The authors have declared that no competing interests exist.

premature neonates delivered before ~35 weeks of gestation often lack lung surfactant and suffer from neonatal respiratory distress syndrome (NRDS) [2]. Surfactant replacement therapy (SRT) with exogenous surfactant substantially decreases NRDS fatalities by providing LS until the babies make enough of their own endogenous surfactant [3]. In another example of the importance of LS, acute respiratory distress syndrome (ARDS), with a mortality rate of ~40% [4], results from many forms of insult to the lungs involving inflammation and alveolar injury that impair LS function. Furthermore, severe acute respiratory syndrome coronavirus-2 (SARS-CoV-2, also known as COVID-19) replicates inside alveolar type-2 (AT2) cells, the only cells that can synthesize, secrete, and recycle all components of LS. In so doing, COVID-19 damages AT2 cells and decreases production of LS, which manifests in ARDS [5,6]. Not only is LS thus essential for life, but SP-B itself is an essential component of LS.

The lipids and proteins in LS work in a concerted fashion; both components are required for surfactant function. About 90% of LS by mass are lipids. Glycerophospholipids predominate, constituting roughly 80% of the lipid portion [7] among which, phosphatidylcholine (PC) contributes the most, 60–70% by mass [8]. Unlike cell membranes, LS contains ~50% dipalmitoyl PC (DPPC). The high content of DPPC offers lung surfactant distinctive properties based on DPPC's ability to pack tightly and thus dramatically reduce surface tension [9,10]. The remaining ~5–10% of LS is comprised of the four surfactant proteins (SPs)-A, B, C, and D. SP-A and SP-D are large, water-soluble lectins important to innate immunity response against inhaled pathogens [11,12]. SP-B and SP-C are small hydrophobic proteins that facilitate surface activity of LS lipids [13]. SP-C and, especially SP-B, have crucial roles in respiration via their roles in promoting lipid structures important in 1) surfactant adsorption to the air-water interface; 2) the formation of the requisite 3D structures when LS material comes out of the air/water interface upon compression of the surface during expiration; and 3) efficient re-insertion of LS into the air-water interface upon expansion during inspiration [10]. Transgenic mice that do not produce SP-B and human infants with congenital genetic abnormalities that prevent them from making SP-B do not survive [14–16]. Conversely, SP-C-deprived mice survive into adulthood; although they develop lung-related complications later in life [17,18]. SP-B is thus an indispensable component of LS.

SP-B is encoded by the *SFTPB* gene on chromosome 2 as a pre-pro-protein of 381 amino acids. The mature form of SP-B has 79 amino acids and a molecular mass of 8.7 kDa. The protein is mostly found as a covalent homodimer when extracted from animal sources with classical methods that use organic solvents, although higher order assemblies have also been reported [19–22]. The primary structure of mature SP-B is highly cationic, with an overall charge of +7 per chain, derived from nine positive and two negatively charged amino acids [13]. SP-B conserves the six cysteine residues and certain hydrophobic residues of saposins making it a member of the saposin-like protein (SAPLIP) superfamily, a large group of proteins with diverse lipid-related functions [23,24]. Based on homology modelling with SAPLIPs, SP-B is expected to take on a structure with four to five helices and

three intrachain disulfide bridges (green lines in Fig 1A) [25,26]. Unlike other saposin superfamily proteins, SP-B possesses an extra cysteine unique to SP-B at Cys48 [27], which forms a disulfide bond with Cys48 on another SP-B molecule to make a disulphide-linked homodimer [26,28,29]. A potential salt bridge between E51 and R52 is also frequently postulated to help stabilize oligomeric structures of SP-B [29].

While no atomic-resolution structure of SP-B has been determined, there are some important clues to SP-B's potential conformations found in the experimental structures of other SAPLIPs. SAPLIP structures fall into two main forms, closed (Fig 1B) and open (Fig 1C). Given that the structure of LS changes dynamically with each breath, it is not too much of a stretch to consider that SP-B's structure might convert between open and closed conformations as part of its various lipid-restructuring activities. Thus far, studies of SP-B have been limited to constructs with 34 (Mini-B), 41 (Super Mini-B) or smaller fragments of SP-B's 79 residues that can be produced by chemical synthesis [32,33], with SP-B obtained from lavaging animal lungs, or most recently, with a recombinant construct that consists of about half of SP-B's residues fused to SP-C [34]. The characteristics of the SP-B obtained from animals depend very much on if SP-B is extracted with or without the use of organic solvents [21]. In 2015, Olmeda *et al* showed that when SP-B was solubilized from LS using the detergent CHAPS (3-((3-cholamidopropyl) dimethylammonio)-1-propanesulfonate) in place of organic solvent and then reconstituted into lipids, it formed large oligomeric structures including strings and rings. These authors and others [22,35,36] suggest that the rings could facilitate lipid flow between LS lipid structures, promote tight packing between adjacent membranes, and also switch overall conformation as the individual SP-B units switched between open and closed conformations.

Despite SP-B's essential role in sustaining life, producing recombinant full length SP-B in the lab has proven to be an unsolved problem. The absence of a method to make full length SP-B recombinantly presents a major challenge in basic research aimed at understanding the structure-function relationships that underlie SP-B's essential contributions to lung function, as well as in the application of SP-B containing surfactants in the clinic where there are dire needs. Without the ability to make SP-B with point mutants, or chimeric SP-Bs, or many of the usual tools employed to determine the structure-function relationships of a protein, our basic understanding of SP-B's mechanisms have been sorely lacking. On the clinical side, while exogenous lung surfactant therapy for neonates has been life-saving, these surfactants have traditionally come from animal lung lavages [37], which has shortcomings for neonates and has also not supported effective therapy for larger patients with ARDS from COVID or other adverse events. In neonates with NRDS, the cost of animal-derived surfactant is high enough to pose challenges in resource-poor countries [38] and there are also concerns with potential allergic reactions and religious objections, depending on the animal source. Successful application of SRT in adults with ARDS has been challenged by 1) the requirement for the hydrophobic protein component of LS for optimal

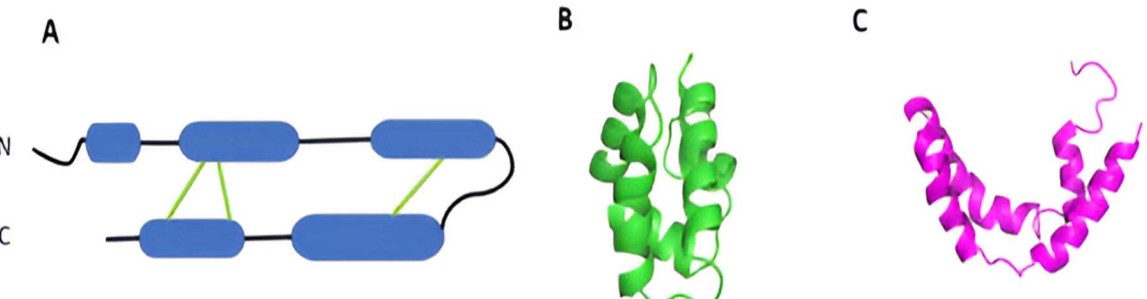

**Fig 1. The secondary structure of SP-B and selected saposin-like proteins.** Panel A: the predicted secondary structure of SP-B with disulfides indicated by green lines. Panel B: the solution structure of NK-lysin (1NKL) adopting the closed conformation [30]. Panel C: solution structure of human saposin C (1SN6) adopting the V-shaped, open orientation [31].

SRT function [39,40]; 2) the difficulty of procuring sufficient animal-sourced (and thus protein-containing) LS for adult-sized patients who require larger and repeated SRT doses; and 3) the limited scope for modifying animal-sourced SRT to resist the inactivating condition [10,41,42] in the lungs of ARDS patients. Furthermore, there is currently much interest in using LS for drug delivery [43–46], for which the ability to produce synthetic SP-B could be important.

Although long stretches of the mature protein have been synthesized and show excellent activity, to our knowledge, there is nothing in the published literature demonstrating production of the mature form of SP-B either by recombinant expression, cell free expression, or chemical synthesis. The absence of a method to make a protein that is so badly needed for research and clinical use likely relates mainly to SP-B's extreme hydrophobicity, as exemplified by its exceptionally high GRAVY index of over 1 [24]. SP-B's hydrophobicity is not only high compared to other saposin family members, which range from about -0.6 to 0.4, but also high compared to transmembrane protein domains, which typically have a GRAVY index close to 0 and are almost never higher than 0.7 [47]. In lungs, SP-B is expressed by alveolar type II cells first as a pre-pro-protein with much more hydrophilic saposin-like domains both N- and C-terminal to the mature SP-B sequence [48,49]. The reduction in overall hydrophobicity offered by the pro-domains has made it possible to express the N-terminal pro-domain by itself [50] and to express SP-B flanked by the N-terminal pro-domain [51] (although not in the absence of the pro-domain). The SP-B protein presented in this work, unlike in [51], is not tethered to the pro-domain. It does lack the N-terminal 7 residues of the native sequence, however preliminary work shows that it has similar secondary structure to full length [52] SP-B, as well as surface activity as measured by Langmuir surface balance and it is much more like the native sequence than peptides that have been explored for use in the clinic like KL4 [32] and Mini-B [33,53]. The work presented here represents the culmination of over 10 years of effort to find an expression construct, *Escherichia coli* strain, purification protocol and refolding protocol that would result in the production of functional, mature, human SP-B.

## 2. Results

### 2.1. Design of the expressed SP-B construct

The expression construct design for SP-B (Fig 2) is based on that previously used for SP-C [54], which employed *Staphylococcus* nuclease A (SN) as an N-terminal fusion protein to help increase solubility, reduce toxicity, promote folding and decrease proteolysis [55,56]. In place of the original thrombin enzyme cleavage site between SN and SP-C, we instead inserted a chemical cyanogen bromide (CNBr) cleavage site to allow cleavage at methionine residues under denaturing conditions. To avoid cleavage of the protein products themselves, two SP-B methionine residues and those in SN were mutated to leucine (underlined in Fig 2B). As suggested by Sharifahmadian *et al.* [57], the N-terminal insertion sequence (FPIPLPY) found in wild type SP-B was removed to improve the handleability of the recombinant SP-B. Moreover, to reduce complications from potential formation of non-native disulfide bonds, we have mutated Cys 48, the residue responsible for homo-dimer formation, to Ser. LS from mice with the Cys48Ser mutation shows altered LS activity [58] although the mice remain healthy, perhaps, as has been suggested, because SP-B dimerization can also occur via non-covalent interactions [59]. The recombinant protein is referred to subsequently as rSP-B ($\Delta7_{NT}$C48S-SP-B-6His). It is important to note that we have also produced a recombinant SP-B that includes the N-terminal insertion sequence, but it yields less protein and we have not performed extensive structure-function characterization on it, choosing instead to work initially with the higher-expressing $\Delta7_{NT}$ protein presented here.

### 2.2. rSP-B ($\Delta7_{NT}$C48S-SP-B-6His) expression

The rSP-B ($\Delta7_{NT}$C48S-SP-B-6His) expression plasmid was transformed into many *E. coli* strains including BL21, C41 and Origami, but only showed detectable expression in C43 (DE3) cells [61]. The data in Fig 3 are consistent with the protein construct expressing mainly in inclusion bodies, i.e., the portion of cell lysate solubilized with a solution containing both

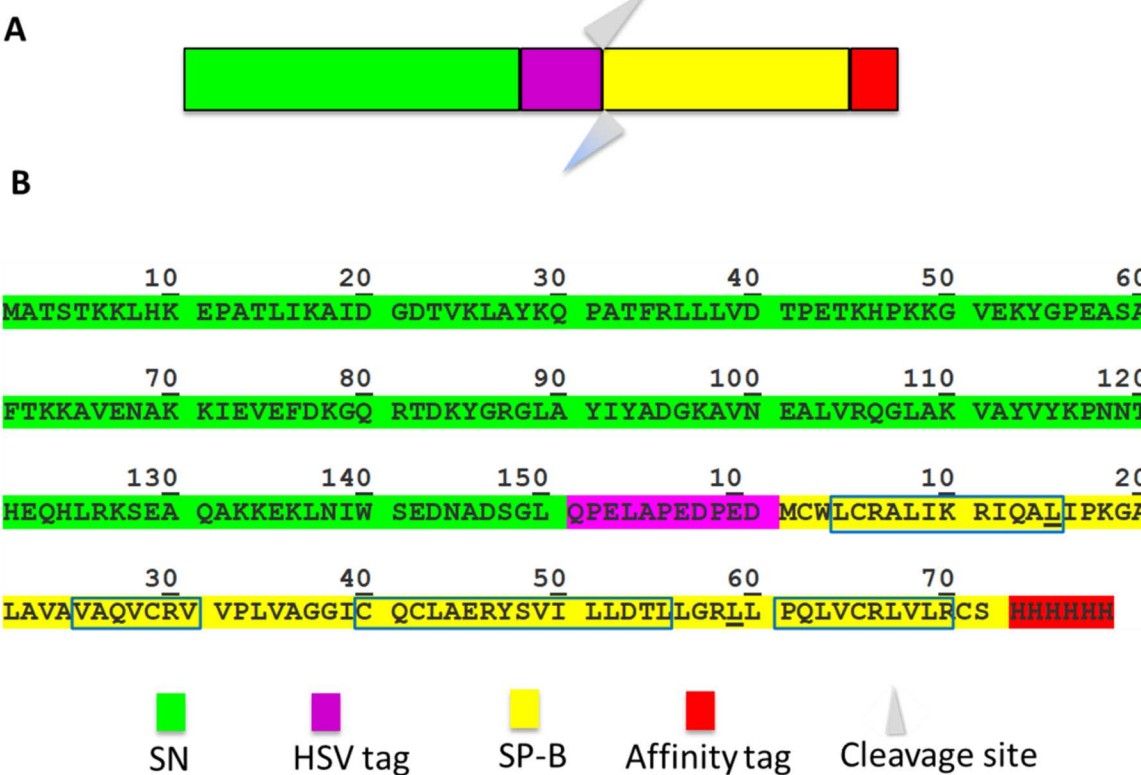

**Fig 2. The expression construct with the SN fusion partner (green), the HSV tag (magenta), rSP-B (Δ7_NTC48S-SP-B) (yellow), and the 6XHis affinity tag (red).** (A) Schematic of the vector construct. (B) Amino acid sequence of the protein construct. The two underlined leucine within SP-B's sequence are point mutants (methionine to leucine). The boxes within rSP-B's sequence indicate the residues that are predicted to fold into α-helices [60].

urea and the detergent CHAPS. The fusion protein is not apparent in the Coomassie stained gel (Fig 3A) as its expression level is too low to allow it to be visualized against the background of *E. coli* proteins. However, recombinant SP-B expression was confirmed with a Western blot of a duplicate gel probed with anti-SP-B antibody (Fig 3B, red arrow) which shows the most intense band at ~30 kDa, close to the expected mass for the fusion protein, i.e., 26.5 kDa. It is likely that the exceptionally hydrophobic nature of SP-B and/or the presence of CHAPS account for the difference in apparent MW on the gel and the expected MW for the construct. Hydrophobic proteins, especially membrane-associated ones, often exhibit anomalous migration in SDS-PAGE due to their interactions with detergents and the presence of disulfide bonds, which can alter migration patterns [62,63]. Some less intensely antibody positive bands are present in all three portions of the lysate at lower MWs, roughly consistent in size with Δ7_NTC48S-SP-B alone; these have been observed in multiple preparations of the protein and we speculate they may represent Δ7_NTC48S-SP-B that has been cleaved from SN by bacterial proteases that can act non-specifically [64,65].

## 2.3. rSP-B (Δ7_NTC48S-SP-B-6His) purification

Purification of the protein is done via a two-stage immobilized metal ion affinity chromatography (IMAC) scheme. The initial IMAC separates the expressed protein construct from the rest of bacterial cell lysate. Buffers were maintained at pH 7.4 during both stages of purification to ensure optimal binding of the 6xHis-tagged rSP-B (Δ7_NTC48S-SP-B-6His) to the resin. Δ7_NTC48S-SP-B-6His is then chemically cleaved from SN with CNBr, and the digest subjected to a second

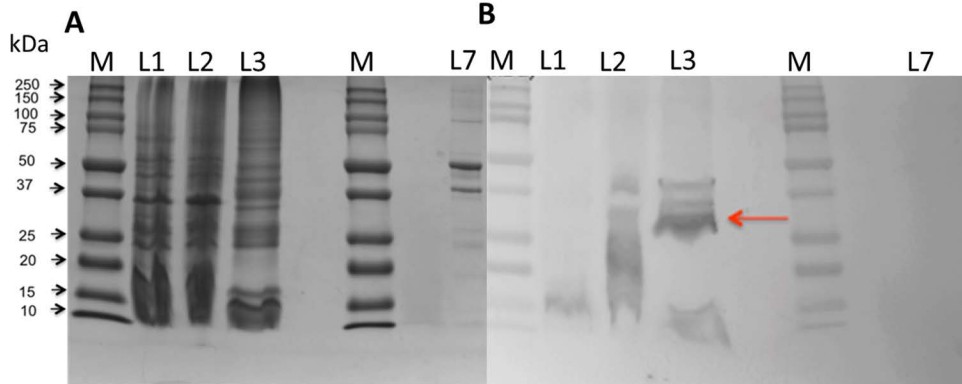

**Fig 3. rSP-B (Δ7$_{NT}$C48S-SP-B-6His) expresses mainly in inclusion bodies.** (A) Non-reducing 12% Tris-Glycine SDS-PAGE of mechanically lysed cells before purification. L1: Tris-buffered saline soluble portion of the cell lysate. L2: 0.2% (w/v) CHAPS soluble fraction of the pellet from L1. L3: 0.2% (w/v) CHAPS + 6 M urea soluble fraction of the pellet from L2. L7: 40 μl of bovine serum albumin (BSA) (negative control for Western in panel B). The gel was stained with Coomassie Brilliant Blue R-250. (B) Duplicate gel of A probed with polyclonal anti-SP-B rabbit antibody (Seven Hills Bio-reagents, Cincinnati, OH). L7 loaded with BSA as negative control does not show any reaction. See Section 5.3 for further details.

IMAC column. Since Δ7$_{NT}$C48S-SP-B-6His retains the 6XHis affinity tag, it remains bound to the 2$^{nd}$ IMAC while the fusion partner, SN, flows through. The recombinant SP-B (Δ7$_{NT}$C48S-SP-B-6His) is then folded while immobilized on the nickel column (Section 2.4 below).

SDS-PAGE carried out on the uncleaved protein after the first IMAC column shows a band at the expected molecular weight of 26.5 kDa (Fig 4A, lane 1). As expected, after CNBr cleavage the gel shows bands at the approximate molecular weights for SN, 17.9 kDa, and for rSP-B (Δ7$_{NT}$C48S-SP-B-6His), 8.5 kDa (Fig 4A, lane 2). The 8.5 kDa MW is as expected; even though native SP-B can normally oligomerize via C48, this residue has been mutated in the recombinant protein and the detergents are likely to disrupt non-covalent interactions that might stabilize SP-B oligomers [59].

A Western blot performed after CNBr cleavage confirmed that only the 8.5 kDa band material binds to the monoclonal anti-SP-B mouse antibody (Fig 4B). Thus, rSP-B (Δ7$_{NT}$C48S-SP-B-6His), does behave as expected in that: 1) it elutes from Stage 1 IMAC at higher imidazole concentrations; 2) it exhibits the expected fusion protein MW (near 26.5 kDa); 3)

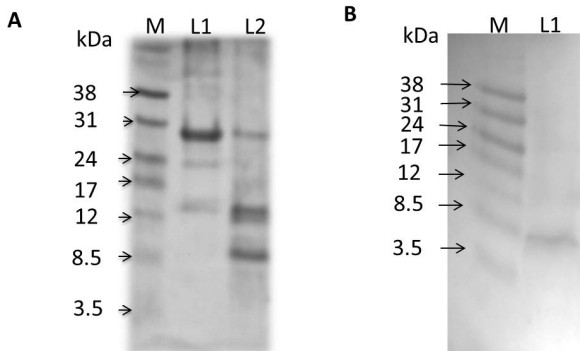

**Fig 4. rSP-B (Δ7$_{NT}$C48S-SP-B-6His) cleavage from SN fusion protein and detection with anti-SP-B antibody.** A) Coomassie-stained non-reducing 16% Tris-tricine SDS-PAGE before (L1) and after (L2) CNBr cleavage of SN from rSP-B (Δ7$_{NT}$C48S-SP-B-6His) B) Immunoblot of CNBr cleaved protein transferred from a separate 16% Tris-Tricine SDS-PAGE. The blot was probed with monoclonal anti-SP-B mouse antibody (Seven Hills Bioreagents, Cincinnati, OH) (L1) and detected using an anti-mouse secondary antibody and visualized using colorimetric detection.

CNBr cleaves the fusion into two polypeptides of the expected MWs (17.9 kDa and 8.5 kDa) and; 4) only the 8.5 kDa polypeptide stains with a monoclonal anti-SP-B antibody.

## 2.4. Folding and detergent exchange

rSP-B ($\Delta7_{NT}$C48S-SP-B-6His) is folded while immobilized on the IMAC column. As part of this folding process, rSP-B can be exchanged from CHAPS into a variety of lipid or detergent systems. First, the IMAC-immobilized rSP-B is washed with CHAPS in decreasing amounts of urea from 6 M to 0 M. The column is then washed with buffer only, i.e., no CHAPS, followed by washing with the desired lipid or detergent, in this case detergent micelles composed of 9:1 dodecylphosphocholine (DPC): sodium dodecyl sulfate (SDS). We have also exchanged rSP-B into pure DPC and a variety of lipids, including 1,2-dimyristoyl-sn-glycero-3-phosphocholine (DMPC), 1-palmitoyl-2-oleoyl-sn-glycero-3-phosphocholine (POPC), 1-myristoyl-2-hydroxy-sn-glycero-3-phosphoglycerol (LMPG), 2-dipalmitoyl-sn-glycero-3-phosphocholine/1,2-dipalmitoyl-sn-glycero-3-phosphoglycerol (DPPC/DPPG, 4:1), DPPC/1-palmitoyl-2-oleoyl-sn-glycero-3-phosphoglycerol (POPG) (7:3), and DPPC/DPPG/1-palmitoyl-2-oleoyl-sn-glycero-3-phosphoglycerol (POPG) (50:25:15) to better simulate the lipid environment found in lung surfactant. Finally, the protein is eluted from the column with imidazole or by reducing the pH.

The yield of rSP-B ($\Delta7_{NT}$C48S-SP-B-6His) depends on which detergent or lipid it is exchanged into; more protein is obtained if anionic detergent or lipid is included compared to if only zwitterionic lipid/detergent is used. If lipid is used it is important to reduce the size of the lipid structures, for example by freeze-thawing, before applying to the column. Typical yields are in the range of 0.1 to 0.3 mg purified protein per liter of initial bacterial cell culture. We have obtained the highest yields with 9:1 DPC:SDS.

## 2.5. Secondary structure characterization

The secondary structure of rSP-B ($\Delta7_{NT}$C48S-SP-B-6His) in DPC/SDS (9:1) micelles was probed via circular dichroism (CD) (Fig 5) and shows minima at 208 and 222 nm that are characteristic of α-helical structure. The mean residue ellipticity of rSP-B at 222 nm indicates the folded, recombinant protein is 47% α-helical in 0.2% DPC/SDS (9:1), which is consistent with what is found for SP-B extracted from animal sources. SP-B derived from animal lung lavage and probed in methanol or in lipids typically exhibits slightly less than 50% ellipticity [66,67].

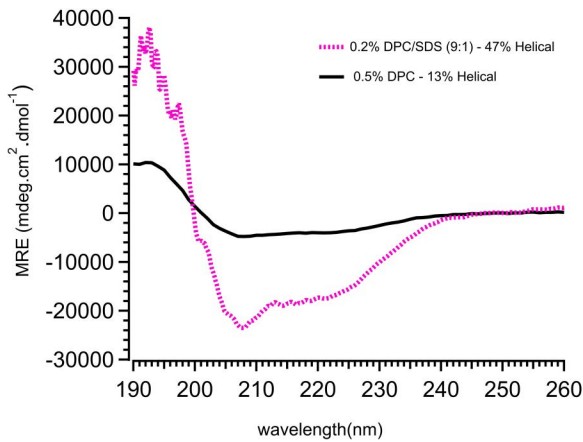

**Fig 5. Mean residue ellipticity (MRE) of rSP-B in different detergent environments.** The α-helical content estimation of rSP-B in 0.2% DPC/SDS (9:1), and 0.5% DPC indicates that rSP-B is most helical in DPC/SDS detergent micelles.

The α-helical content of rSP-B varies significantly across different detergent environments: 47% in 0.2% DPC/SDS (9:1), and 13% in 0.5% DPC (Fig 5). The higher α-helical content in DPC/SDS mixture suggests a mixed zwitterionic and anionic head groups best promotes the formation of α helices in rSP-B. The CD spectra of rSP-B in methanol show that breaking the disulfide bonds with a reducing agent leads to a ~50% decrease in helicity (S1 Fig).

## 2.6. rSP-B (Δ7$_{NT}$C48S-SP-B-6His) surface activity

rSP-B (Δ7$_{NT}$C48S-SP-B-6His) exhibits surface activity, as assessed by Langmuir-Wilhelmy surface balance, which measures the surface pressure of lipids and lipid/protein mixtures at the air-water interface as a function of surface area. The incorporation of 2 wt% of recombinant SP-B in DPPC/DPPG (4:1) film induces an inflection point and plateau in the pressure-area compression isotherm at a surface pressure of ~ 45 mN/m (* in Fig 6A), a feature that is not seen in the absence of protein. Similar features have been attributed to monolayer-to-multilayer transitions facilitated by SP-B [68,69]. Note that with recombinant protein in DPPC:DPPG, the highest surface pressures achieved do not approach 72 mN/m as does native LS, but is limited to ~64 mN/m; high surface pressures were however obtained with recombinant protein in DPPC:POPG after repeated compression cycles (Fig 6B).

Another important characteristic of a good LS is film stability during repeated dynamic breathing cycles. Without protein, DPPC/POPG (7:3) films do not retrace the isotherms on sequential compression/expansion cycles, but do with the inclusion of rSP-B (Δ7$_{NT}$C48S-SP-B-6His) at 2, 4, 6, and 10 wt%, with the best behavior at 6 wt% (Fig 6B). The reincorporation of material into the surface film upon expansion promoted by the recombinant protein is seen in the flattening in the slope [70] at ~ 25 mN/m (arrow in Fig 6B). The behaviour of lipid films containing recombinant SP-B (6 wt%) under cycling is thus encouraging as SP-B has been proposed to be critical in lung surfactant film replenishment during dynamic breathing cycles [71].

## 3. Discussion

This work represents the first report of a recombinant method to produce the essential protein SP-B in the lab and thus enables a wealth of possibilities for future SP-B work. SP-B made recombinantly in bacteria will permit studying structure-function relationships in full-length or near full-length SP-B by enabling the production of point mutants and

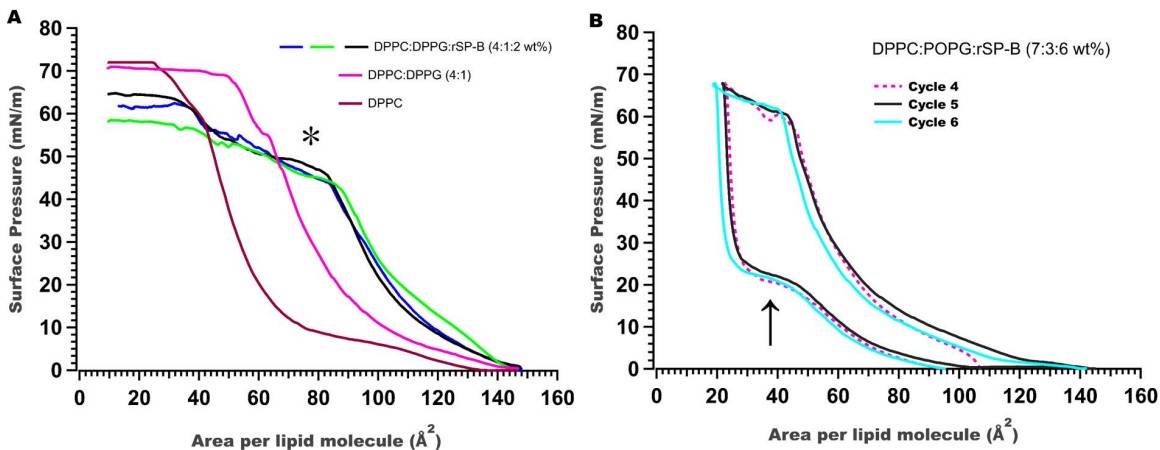

**Fig 6. rSP-B (Δ7NTC48S-SP-B-6His) exhibits surface activity.** A) The pressure-area isotherms of DPPC (brown), DPPC/DPPG (4:1) (red) and DPPC/DPPG with 3 independent preparations of rSP-B (Δ7$_{NT}$C48S-SP-B-6His) (4:1:2 wt%) (blue, green, black). The presence of 2 wt% recombinant protein induces an inflection point at ~45 mN/m (*). B) Retracing of cyclic compression and expansion isotherms of DPPC/POPG/rSP-B (Δ7$_{NT}$C48S-SP-B-6His) (7:3:6 wt%) demonstrates film stability and reincorporation of surface material upon expansion(arrow).

chimeras. Given that proteins made recombinantly in bacteria are now commonly approved by the FDA as therapeutics [72], it should also now be within reach to include SP-B produced this way in novel therapeutics for common and frequently fatal conditions such as ARDS [39,40] or for drug delivery [41–43].

Although rSP-B ($\Delta 7_{NT}$C48S-SP-B-6His) expression was attempted in many strains of *E. coli*, that success was only achieved with C43 cells provides some clues as to how rSP-B behaves in bacterial cells. C43 cells are known to support production of toxic and membrane proteins [73], and it is possible that the presence of additional membrane sequesters at least some recombinant SP-B. That said, rSP-B ($\Delta 7_{NT}$C48S-SP-B-6His) appears to locate to inclusion bodies as well since much of it requires both detergent and a denaturing agent to recover it from the cells (Fig 3).

The position of the 6XHis tag on rSP-B ($\Delta 7_{NT}$C48S-SP-B-6His) is also critical. Although protein expression is successful with the 6XHis-tag on the C-terminus of the recombinant protein construct, when the 6XHis tag is at the N-terminal end of the construct, there is little detectible expression – either because the protein does not express well or does not bind the IMAC column well enough to allow detection of the protein product. The C-terminal 6XHis tag remains on rSP-B ($\Delta 7_{NT}$C48S-SP-B-6His) after it is purified and thus it is important to consider its potential effects on SP-B structure and function. Given the experiments are done at a pH well above histidine's pKa, the tag will likely not much alter the overall charge of the protein. Histidine's amphipathic nature and propensity for membrane polar/apolar interface [74,75] may affect rSP-B's membrane localization and surface activity, warranting further investigation in future studies. To address this, it may be possible to add an additional affinity tag to the expressed protein, facilitating binding to the column during folding while allowing the C-terminal His-tag to be removed, thereby minimizing its potential impact on surface activity.

rSP-B ($\Delta 7_{NT}$C48S-SP-B-6His) contains 6 cysteine residues that in the native SP-B structure form 3 intra-chain bonds [27]. Protein purification is carried out under normal benchtop conditions in the absence of reducing agents and thus the disulfide bonds are expected to oxidize and form disulfides as the purification proceeds. Previous work in our lab with the peptide of SP-B called Mini-B that contains two disulfide bonds [76] has indicated that this oxidation occurs during purification and that the bonds form correctly (as determined by NMR structure determination) so long as the polypeptide is in an appropriate environment such as anionic-species containing liposomes or detergent micelles [77]. To assess if rSP-B ($\Delta 7_{NT}$C48S-SP-B-6His) forms native disulfides we have attempted to enzymatically cleave it, which would have enabled us to assess the disulfide bond formation pattern by mass spectrometry. Unfortunately, we have not found conditions where the recombinant protein is soluble enough to be accessible to proteases but that also allow the protease to be enzymatically active, even for proteases that tolerate somewhat denaturing conditions. However, we strongly suspect that the intra-chain disulfides are forming correctly because the conserved hydrophobic residues [24] would guide the disulfide formation and because rSP-B spends most of its time bound to the resin preventing incorrect inter-chain disulfide bonds from forming. This is backed up by the observation that rSP-B ($\Delta 7_{NT}$C48S-SP-B-6His) forms the correct secondary structure (Fig 5) and exhibits interfacial activity in functional assays employing Langmuir trough measurements (Fig 6) [52] although more extensive functional testing is desirable as part of future work.

The structural stability and helical content of rSP-B are crucial for its role in surfactant function within the lungs. Higher helicity could enhance its ability to interact with lipid membranes, which is essential for surfactant activity. The similarity in helical content between natural SP-B and rSP-B in the DPC/SDS mixture indicates that this detergent environment closely mimics the native conditions of the protein in the lung surfactant (Fig 5) [66,78], and thus underscores the relevance of DPC/SDS system for studying the structural and functional aspects of SP-B.

## 4. Conclusions

In summary, producing recombinant SP-B in the lab is a major step forward for lung surfactant basic research and clinical applications. To this point, SP-B research has been held back by this absence, although SP-B is essential for life and a key component of lung surfactant and of exogenous therapeutics for common and life-threatening conditions like ARDS. The work opens up new possibilities to study key structure-function determinants of SP-B via the many techniques that

require recombinant protein and provides a means to produce SP-B on the scale needed for inclusion in exogenous surfactants badly needed for conditions like COVID- and non-COVID-related ARDS.

## 5. Materials and methods

### 5.1. Transformation

The plasmid (Genscript), 1 μl of the vector DNA, i.e., the pET11a *E. coli* T7 expression plasmid that contained the insert, rSP-B ($\Delta7_{NT}$C48S-SP-B-6His) was transformed into *E. coli* C43 chemically competent cells (VWR International, Edmonton, Canada). Glycerol (1:1 v/v) stocks were created from a single colony from a 2XYT (16 g/L tryptone, 10 g/L yeast extract and 5 g/L sodium chloride) plus ampicillin (1 μg/ml, Fisher BioReagents) agar plate and stored at -80°C.

### 5.2. Expression

Cells were grown at 37°C from overnight stocks in 2XYT containing ampicillin (final concentration 0.05 mg/ml) until an optical density (OD600 nm) of 0.6 was reached. Expression was induced by adding isopropyl β-D-1-thiogalactopyranoside (IPTG) (Gold Biotechnology) to a final concentration of 0.4 mM/L. Following induction, the culture was incubated at 37°C for 3 hours in a total culture volume of 6 L (1 L per shake flask). Cells were then harvested by centrifugation and resuspended in 20 ml of TBS (50 mM Tris, 150 mM NaCl, pH 7.4) containing 5 mM PMSF (phenylmethylsulfonylfluoride) [79]. Typical cell pellet volumes were 20 ml.

### 5.3. Cell lysis and inclusion body solubilization

Cells were lysed via French pressure (hydraulic pressure liquid shearing force, 10,000 psi) at 4°C in three consecutive cycles or by sonication on ice (four cycles of 20 seconds each). Since rSP-B ($\Delta7_{NT}$C48S-SP-B-6His) expresses mainly as inclusion bodies, 6M urea and 0.2% (w/v) 3-[(3-Cholamidopropyl) dimethylammonio]-1-propanesulfonate (CHAPS, AG Scientific) were added to the lysed cells. The lysate/urea/CHAPS mixture was agitated for 2 hours at room temperature on an orbital shaker before purification as described in Section 5.4 below.

Alternately, for the SDS-PAGE gels shown in Fig 3, the lysate was fractionated and analyzed as follows. After lysis, the lysate was pelleted by centrifugation and the supernatant retained to run on lane 1. The pellet was treated with 0.2% (w/v) CHAPS and centrifuged again. The supernatant, containing the CHAPS-soluble protein, was retained to run on lane 2. The pellet was treated with 0.2% CHAPS (w/v) in 6 M urea and run on lane 3. One gel was stained with Coomassie Brilliant Blue R-250 for one hour and de-stained until the desired bands were observed (Fig 3A). The second gel was used for transfer for Western blotting using polyclonal anti-SP-B rabbit primary antibody (Seven Hills Bio-reagents, Cincinnati, OH) (Fig 3B). The protein molecular-weight size standard used was Kaleidoscope Precision Plus Protein$^{TM}$ (Bio-Rad).

### 5.4. First stage purification

To remove residual DNA from the sample, 10 ml of Ion-Sep DE52, pre-swollen DEAE (diethyl amino ethyl) cellulose (Biophoretics, Sparks, NV) was added to the lysate containing urea and CHAPS. The mixture was incubated on a rotary shaker at room temperature for at least an hour and then centrifuged at 7580 g at 25°C for 15 minutes, retaining the supernatant. This supernatant, containing the fusion protein, was applied to 5 mL of IMAC Sepharose Fast Flow resin (GE Healthcare, UK) that had been pre-equilibrated with binding buffer A (5 mM imidazole, 6 M urea, 0.2% (w/v) CHAPS in TBS, pH 7.4) [80]. The mixture was then transferred to a 50 mL centrifuge tube and gently agitated overnight at room temperature to allow binding of rSP-B ($\Delta7_{NT}$C48S-SP-B-6His) to the resin. The resin was loaded onto a PD-10 column and washed, followed by elution with 10 ml aliquots of an imidazole gradient (5-, 10-, 20-, 50-, 100-, 200- and 300-mM imidazole in binding buffer A). The pH of the buffers was consistently maintained at 7.4 throughout the process.

To remove the excess CHAPS, salt, imidazole and urea, the pooled sample was dialyzed using Spectra POR-4 12,000 to 14,000 molecular weight cut-off (MWCO) dialysis tubing (VWR International, Edmonton Canada) against 3 liters of distilled water overnight in a cold room (+4°C) while stirring. The dialyzed sample was flash frozen using liquid nitrogen and lyophilized overnight (Labconco freeze drier, Kansas City, MO, USA).

Protein concentration measurements showed a loss of ~10–15% of protein during dialysis. A Bradford assay was used for protein concentration determination, as it provided the most accurate results compared to fluorescamine, Biuret, BCA, and absorbance at 280 nm.

## 5.5. Cleavage of *SN* fusion protein from rSP-B (Δ7$_{NT}$C48S-SP-B-6His)

The lyophilized protein sample was resuspended in a 2 ml volume of 70% formic acid. A few small crystals of cyanogen bromide (CNBr 97%, Acros Organics) on the tip of a small spatula, were added to the sample [81]. The vial was wrapped in aluminum foil and was left for 1 to 2 days at room temperature. The reaction was stopped by diluting the sample with distilled water (1:15 dilution). The solution was flash frozen and lyophilized (typical mass after lyophilization was 9 mg). The resulting lyophilisate was resuspended in 5 ml of binding buffer B (6 M urea, 0.5% (w/v) CHAPS, TBS, 5 mM imidazole, pH 7.4) to prepare it for second stage IMAC purification as described in Section 5.6 below.

## 5.6. Second stage purification, folding, and detergent exchange

In the second purification stage, an additional IMAC column was used to separate rSP-B (Δ7$_{NT}$C48S-SP-B-6His) from SN, to fold the protein on the column, and to exchange it into the desired detergent/lipid. The CNBr digested sample from Section 5.5 was incubated with 5 ml of IMAC resin previously pre-equilibrated with binding buffer B. This was left to rotate on a rotary shaker overnight at room temperature in the PD-10 column. The next day, the flow through from the column, containing SN, was collected and the column was washed with 15 ml of the binding buffer B. The rSP-B [82] Δ7$_{NT}$C48S-SP-B-6His) was renatured while still bound to the column by decreasing the concentration of urea in the column. This was done by washing the column with 10 ml of washing buffer (6 M urea, 0.5% CHAPS, TBS and 5 mM imidazole) in stepwise 1 M decrements of urea from 6 M to 0 M [82].

To remove the CHAPS, the column was then washed with 30 ml of 20 mM Tris-HCl buffer (pH 7.4). Following this, detergent exchange was performed by washing the resin with 3 column volumes (15 ml) of the desired detergent or lipid in Tris-HCl buffer (pH 7.4). In case of lipids, to prevent column clogging, the lipids were pre-treated with five cycles of freezing and thawing at 50°C, which is expected to reduce the size of the lipid vesicles [83–85]. Finally, rSP-B (Δ7$_{NT}$C48S-SP-B-6His) was eluted from the column by washing it with 5 column volumes of the elution buffer, either with increasing imidazole concentration (as for first stage IMAC) or with decreased pH (pH 5).

## 5.7. Circular dichroism spectroscopy

For the CD spectra shown in Fig 5, rSP-B (Δ7$_{NT}$C48S-SP-B-6His) was exchanged into 9:1 DPC:SDS (0.2%) as described in Section 5.6. Twenty spectra were collected in milli-degrees at a rate of 1 nm per second using a Jasco J-810 spectropolarimeter (Jasco Inc., Easton, MD) in the far ultraviolet range (190–260 nm) with a 0.5 mm quartz cuvette at 25 °C. The pitch of the scan was set to 1 nm. Spectra were corrected by subtracting the buffer only scan [86] and then the units converted to mean-residue ellipticity (MRE) according to $MRE = \theta/(c \times l \times Nr)$, where $\theta$ is the recorded ellipticity, $c$ is the peptide concentration in dmol/L determined as described in Section 5.4, $l$ is the cell path-length in cm, and $Nr$ is the number of residues in the peptide. Percent α-helicity was estimated from the normalized CD spectra as in [87]. In addition, the effect of disulfide bond reduction on the secondary structure of rSP-B was evaluated by adding a reducing agent. This treatment, shown in S1 Fig, led to a ~50% decrease in α-helicity, as indicated by the CD spectra.

## 5.8. Surface activity measurement

Surface pressure-area (Π-A) isotherms were recorded using a Kibron MicroTrough XS (59 mm X 208 mm, Helsinki, Finland) balance controlled by software version 3.6.1 [88]. Lipids were mixed by weight to a final concentration of 1 mg/ml in a chloroform/methanol solution. Aliquots of rSP-B (Δ7NT C48S-SP-B-6His) at 2, 4, 6, 8, and 10 wt% (with respect to lipids) in a 1:1 (v/v) chloroform/methanol mixture were added to the 1 mg/mL lipid solution. A 10 μl aliquot of the lipid-protein mixture was drop-wise spread onto ultrapure water (18.2 MΩ.cm) from the air-side using a microsyringe (Hamilton, NV, USA). A minimum of ten minutes was given for the organic solvent to evaporate and the film to stabilize. Three ∏-A compression isotherm measurements were recorded and averaged. For the cyclic isotherm, a single continuous measurement was performed, capturing both the compression and relaxation phases over multiple cycles, all conducted at ambient temperature.

## Supporting information

**S1 Fig. The CD spectra of rSP-B in methanol and increasing concentrations of reducing agent Tris 2-carboxyethyl phosphine (TCEP).** The signal at 222 nm, which is sensitive to α-helical structure decreases with increasing TCEP. The spectra have been normalized by relative protein concentration.
(DOCX)

**S2 Data. Raw circular dichroism data for rSP-B in DPC.**
(TXT)

**S3 Data. Raw circular dichroism data for rSP-B in DPC/SDS.**
(TXT)

**S4 Data. Langmuir trough compression isotherms for rSP-B in lipids.**
(XLSX)

**S5 Data. Cyclic isotherms for rSp-B in DPPC/POPG.**
(XLSX)

**S6 Data. Raw, uncropped images for gels and blots in Figs 3 and 4.**
(PDF)

## Author contributions

**Conceptualization:** Valerie Booth.

**Funding acquisition:** Valerie Booth.

**Investigation:** Tadiwos Asrat, Donna Jackman.

**Supervision:** Donna Jackman, Valerie Booth.

**Writing – original draft:** Tadiwos Asrat.

**Writing – review & editing:** Donna Jackman, Valerie Booth.

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
