## [Decision Letter · Decision Letter 0]

14 Nov 2024

PONE-D-24-45785Bacterial expression, purification and refolding of exceptionally hydrophobic and essential protein: Surfactant Protein-B (SP-B)PLOS ONE

Dear Dr. Booth,

Thank you for submitting your manuscript to PLOS ONE. After careful consideration, we feel that it has merit but does not fully meet PLOS ONE’s publication criteria as it currently stands. Therefore, we invite you to submit a revised version of the manuscript that addresses the points raised during the review process.

We look forward to receiving your revised manuscript.

Kind regards,

Michael Massiah

Academic Editor

PLOS ONE

“This work was funded by a National Sciences and Engineering Research Council of Canada Discovery Grant to VB (204115).”

“This work was funded by a National Sciences and Engineering Research Council of Canada Discovery Grant to VB (204115).”

“This work was funded by a National Sciences and Engineering Research Council of Canada Discovery Grant to VB (204115).”

Reviewers' comments:

Reviewer's Responses to Questions

**Comments to the Author**

1. Is the manuscript technically sound, and do the data support the conclusions?

Reviewer #1: Yes

Reviewer #2: Partly

2. Has the statistical analysis been performed appropriately and rigorously? 

Reviewer #1: N/A

Reviewer #2: N/A

3. Have the authors made all data underlying the findings in their manuscript fully available?

Reviewer #1: Yes

Reviewer #2: No

4. Is the manuscript presented in an intelligible fashion and written in standard English?

Reviewer #1: Yes

Reviewer #2: Yes

5. Review Comments to the Author

Reviewer #1: Asrat et al. present a well-written description of a promising recombinant production methodology for surfactant protein-B (SP-B). As is well described by the authors, this protein is critical for life (at least in [some?] organisms with lungs) and the lack of a robust method to produce it either recombinantly or synthetically is a limiting factor for both in-depth studies and for therapeutic application. The authors clearly describe and demonstrate a method to produce SP-B (noting that the final construct lacks a 7-residue segment at its N-terminus, has two Met to Leu substitutions, and one Cys to Ser substitution) as a fusion protein with Staphylococcus nuclease A (SN) and with a C-terminal His6-tag. This fusion protein was demonstrated to be targeted to inclusion bodies upon overexpression in the C43 strain of E. coli, enabling relatively straightforward isolation of the protein. CNBr cleavage was then employed to remove the SN tag, followed by Ni2+-IMAC in order to obtain pure recombinant SP-B. The resulting protein structure was investigated by far-UV CD spectroscopy and its surface activity through LB trough evaluation. As a whole, this is a valuable methodology and should be highly useful. I do however feel that there are several points that it would be beneficial to address prior to publication:

1) In the abstract and again in the discussion, it is noted that C43 cells “overproduce intracellular membranes”. I am not sure that this is a fair generalization to make, as this strain was identified by the Walker group as being generally amenable to toxic/membrane protein production while the citation in question is a quite specific case. Perhaps the SP-B fusion construct also encourages or benefits from intracellular membrane production in C43, but I would prefer to see this noted in a more hypothetical manner given that there is limited evidence in the literature for this as a general phenomenon.

2) The inclusion of a C-terminal His-tag may indeed be perturbing to function, but is not rationalized/discussed until the second-last para. of the Discussion. Are there any literature examples of SP-B-dervied proteins/peptides that retain a His-tag and also retain full surface activity? Beyond the lack of charge argument that is advanced on p. 15 on the basis of the pH being used (I presume pH 7.4 – this could be a little clearer in the methods), is there any potential concern that the aromaticity of the His may come into play with, for example, snorkeling type behaviour that is exhibited with aromatic residues? Although it goes beyond the scope of the present study, would a potentially good strategy be to include an additional Met before the His tag to allow its cleavage and then perhaps an N-terminal His-tag for the SN tag in order to perform reverse purification? Overall, it may not be a concern that there is a C-terminal His6-tag, but I would prefer to see this feature of the recombinant SP-B construct be discussed more broadly through the manuscript (especially making this clear in the results section).

3) Given the importance of S-S bonds in SP-B, I was a bit surprised that there was no discussion of oxidation during refolding until the final para. of the Discussion. It seems that this is being anticipated to take place in concert with refolding on the Ni2+ column? What conditions are being used in that case, and isn’t the pH applied sub-optimal to favour disulfide formation? (Or, is this mirroring the conditions used to refold Mini-B with concomitant disulfide formation?) While it is noted that the bonded state is difficult to assess, would it be feasible to use either near-UV CD (especially comparing the appropriate region of the spectrum for the purified protein without reducing agent vs. following reduction) or comparing SDS-PAGE under non-reducing and reducing conditions to go beyond the inference that the near-full-length SP-B should behave similarly to Mini-B?

4) Relating to (3), what is known about the necessity or lack thereof of the S-S induced dimerization of SP-B? (i.e., is there concern about the Cys to Ser substitution that was incorporated in terms of functional applicability of this recombinant protein?)

5) When carrying out far-UV CD, it’s not fully clear how the protein concentration was determined, nor why the particular methodology chosen was used to infer helical content. Given that there are, for example, deconvolution datasets specific to membrane proteins (recognizing that SP-B isn’t an integral membrane protein), how different might those results be? How does the percentage of helicity inferred from evaluating normalized CD spectra compare to the expectations based on secondary structure predictions shown in the schematic figures? And could the dramatic differences in helical percentage in different environments be an artifact of the helical estimation methodology used rather than a true representation of differences in helicity in these environments? If the disulfide bonds are indeed formed (point 3) it seems hard to rationalize such extreme changes in helical content.

6) Based on the speculation that bacterial proteases are leading to products of the approximate MW of the SP-B, are there any insights from the sequence of the fusion protein as to which protease(s) might be doing so and what the cleavage product would be?

7) The figure caption for Fig. 4 (B) is a little unclear (i.e., what is the “different” gel that is referred to? How was the monoclonal antibody detected?)

8) It’s unclear what’s meant by “only the near 8.5 kDa band material binds to the monoclonal anti-SP-B mouse antibody”.

9) The authors consistently refer to the protein as showing “some” surface activity. This seems an odd choice of wording and perhaps could be either made more specific in nature or just referred to as being surface active?

Some additional minor comments:

10) It would be good to provide reference(s) for the statement that the anomalous migration by SDS-PAGE is expected (p. 9)

11) Abstract and again on p. 7 – italicize Staphylococcus?

12) Abstract – specify that C43 is an Escherichia coli strain. (See also comment about whether it is suitable to generalize statement finding of intracellular membrane production.)

13) p. 7, para. 1 – correct “presented in this work presents”; spell out Escherichia on first use [here]

14) Fig. 2 caption – capitalize “Amino” at start of (B)

15) p. 13, first para. of discussion – probably more suitable to refer to this as “first report” rather than “first publication”?; second sentence – remove comma after permit?

16) p. 14, 2nd para. – not fully clear how an MD simulation (presumably MD, rather than “molecular simulation”?) “might” show cholesterol binding sites; the entire sentence is worded rather speculatively.

17) Methods are fairly well described, but some additional references for sources and precedents for key protocols would be beneficial to include

18) Section 5.7 – include “spectroscopy” in title? Rate of data collection and pitch? Was buffer/blank correction employed? (“Baseline correction” is mentioned, but this has a different meaning and it’s not fully clear how this would be carried out in CD?)

19) Section 5.8 – Specify measurement being carried out in the section title rather than instrument; the wt% per mg/mL of lipids in a v/v solution is very hard to follow - perhaps this could be described in a different way?; I believe that units for water should be MOhm x cm (not just MOhm)?; It would also be good to clarify what was done with the three measurements that were carried out for each sample (i.e., are these cyclic responses that are considered individually or were these averaged, or some other treatment?)

Reviewer #2: The manuscript describes to my knowledge the first recombinant production and purification of (almost) full-length human SP-B. Results are valuable for basic research on this protein and potentially for other, including therapeutic, applications.

The manuscript has some flaws (1) in the methods section, which is incomplete, and (2) in the results & discussion, where some conclusions are not supported by the results shown and therefore seem too speculative.

Methods:

• Expression (p17): Culture volume? Type of culture (e.g. shake flask)? Wet cell mass would also be helpful to know.

• First stage purification (p18): Is there a centrifugation step missing before adding the IMAC resin? How much IMAC resin was added? Apparently, after batch adsorption, the resin is transferred to a column (what type?), this information is missing. What is the "loading buffer"?

• Dialysis (p18): Transferring a hydrophobic protein into distilled water by dialysis should result in aggregation: was aggregation assessed at that stage, or was turbidity observed? What is the protein yield of the dialysis step? I would expect considerable loss, as precipitated protein might stick to the dialysis tubing. How was protein concentration measured anyway?

• P18, bottom: "pellet" should read "lyophilisate".

• 2nd stage purification (p19). Calling two different buffers "loading buffer" and "second loading buffer" is maybe not the best idea. Please assign unique names to solvents.

• Lipid preparation (p19) "After each cycle of freezing, the lipids were thawed at 50°C to break the large multilamellar vesicles into smaller vesicles". Provide a reference confirming that this method leads to smaller vesicles. To my knowledge, freeze-thaw cycles lead to vesicle fusion, i.e. the opposite of the desired effect.

Results not supported by data

• Folding state: as you call your procedure "refolding", one would assume a native state of the protein. The CD data vary widely in alpha helix content, depending on the environment. Caption of Fig. 5 suggests the protein is near-native in DPC/SDS. However, SDS induces alpha helix in almost all proteins, therefore, this conclusion cannot be taken. I suggest recording a CD spectrum in SDS only as a control.

• In the methods section, numerous lipid/detergent mixtures are given, but for only three of those results are presented. Why? It would be interesting to see how the protein behaves (alpha helix content) in the other mixtures.

• Disulphide bond formation (p15): no evidence of correct disulphide formation in results. You refer to results from related peptide without explaining how disulphide formation was determined there.

• Function (p13): If I understood the procedure correctly, the protein is transferred back to organic solvent before being added to the lipid film. If that is the case, prior "refolding" would be irrelevant. I am aware that this is how SP-B from lung tissue has been tested before. To me it seems that this peptide, due to its small size and overall structure is assumed to "refold" from any starting conformation if it is transferred into a suitable lipid/detergent environment.

• Role of CHAPS (p14): Highly speculative. If true, will e.g. CHAPSO or CHS work as well (specific interaction with steroid structure)? This could be tested easily. In the purification, CHAPS is used below its CMC (~5 mM), meaning that it doesn't form micelles. Also, 6M urea further reduces hydrophobic interactions. Have you tested to solubilize in urea without CHAPS, or, alternatively, in urea with other detergents below their CMC? This could be the reason why the His tag is available in CHAPS, as binding of the detergent to the protein will be weak anyway.

• Oligomer formation (p14): highly speculative, and could be easily tested, e.g. by analytical SEC.

Miscellaneous:

• Resolution of figures is very poor. Hard to read e.g. MWs on PAGE:

• Headers: 2.6: "Surface activity", 5.8: Surface activity measurement

6. PLOS authors have the option to publish the peer review history of their article (what does this mean? ). If published, this will include your full peer review and any attached files.

**Do you want your identity to be public for this peer review?** For information about this choice, including consent withdrawal, please see our Privacy Policy .

Reviewer #1: No

Reviewer #2: No

---

## [Author Response · Author response to Decision Letter 1]

18 Jan 2025

Response to Reviewers

We thank the reviewers and editor for their time and insights regarding this work. We have carefully revised the manuscript in response to the comments and feel it is now a much stronger piece of work thanks to your advice.

Reviewer 1

“Asrat et al. present a well-written description of a promising recombinant production methodology for surfactant protein-B (SP-B). As is well described by the authors, this protein is critical for life (at least in [some?] organisms with lungs) and the lack of a robust method to produce it either recombinantly or synthetically is a limiting factor for both in-depth studies and for therapeutic application. The authors clearly describe and demonstrate a method to produce SP-B (noting that the final construct lacks a 7-residue segment at its N-terminus, has two Met to Leu substitutions, and one Cys to Ser substitution) as a fusion protein with Staphylococcus nuclease A (SN) and with a C-terminal His6-tag. This fusion protein was demonstrated to be targeted to inclusion bodies upon overexpression in the C43 strain of E. coli, enabling relatively straightforward isolation of the protein. CNBr cleavage was then employed to remove the SN tag, followed by Ni2+-IMAC in order to obtain pure recombinant SP-B. The resulting protein structure was investigated by far-UV CD spectroscopy and its surface activity through LB trough evaluation. As a whole, this is a valuable methodology and should be highly useful. I do however feel that there are several points that it would be beneficial to address prior to publication:”

Point1: “In the abstract and again in the discussion, it is noted that C43 cells “overproduce intracellular membranes”. I am not sure that this is a fair generalization to make, as this strain was identified by the Walker group as being generally amenable to toxic/membrane protein production while the citation in question is a quite specific case. Perhaps the SP-B fusion construct also encourages or benefits from intracellular membrane production in C43, but I would prefer to see this noted in a more hypothetical manner given that there is limited evidence in the literature for this as a general phenomenon.”

Response: We have re-worded the abstract and discussion text in line with the reviewer’s suggestion.

Point 2: “The inclusion of a C-terminal His-tag may indeed be perturbing to function, but is not rationalized/discussed until the second-last para. of the Discussion. Are there any literature examples of SP-B-dervied proteins/peptides that retain a His-tag and also retain full surface activity? Beyond the lack of charge argument that is advanced on p. 15 on the basis of the pH being used (I presume pH 7.4 – this could be a little clearer in the methods), is there any potential concern that the aromaticity of the His may come into play with, for example, snorkeling type behaviour that is exhibited with aromatic residues? Although it goes beyond the scope of the present study, would a potentially good strategy be to include an additional Met before the His tag to allow its cleavage and then perhaps an N-terminal His-tag for the SN tag in order to perform reverse purification? Overall, it may not be a concern that there is a C-terminal His6-tag, but I would prefer to see this feature of the recombinant SP-B construct be discussed more broadly through the manuscript (especially making this clear in the results section).”

Response: Thank you for raising the important issue regarding the potential effects of the C-terminal His-tag on rSP-B function. In response, we have clarified the pH in the methods and improved our discussion of this matter in both the Results and Discussion sections. Additionally, we have incorporated into the discussion, as a potential future direction, the reviewer’s helpful suggestion to explore adding an additional tag for help with purification so that the C-terminal his-tag can be cleaved.

Point 3: “Given the importance of S-S bonds in SP-B, I was a bit surprised that there was no discussion of oxidation during refolding until the final para. of the Discussion. It seems that this is being anticipated to take place in concert with refolding on the Ni2+ column? What conditions are being used in that case, and isn’t the pH applied sub-optimal to favour disulfide formation? (Or, is this mirroring the conditions used to refold Mini-B with concomitant disulfide formation?) While it is noted that the bonded state is difficult to assess, would it be feasible to use either near-UV CD (especially comparing the appropriate region of the spectrum for the purified protein without reducing agent vs. following reduction) or comparing SDS-PAGE under non-reducing and reducing conditions to go beyond the inference that the near-full-length SP-B should behave similarly to Mini-B?”

Response: Good point - we should have provided more detail on the disulfides. We conducted CD experiments with and without the reducing agent TCEP (in methanol to ensure it had access to the disulfide bonds). The reducing agent decreased helicity by ~50%, we’ve added a supplementary figure with this data and included additional text in Section 2.5 (Discussion and Methods) to explain it.

Point 4: “Relating to (3), what is known about the necessity or lack thereof of the S-S induced dimerization of SP-B? (i.e., is there concern about the Cys to Ser substitution that was incorporated in terms of functional applicability of this recombinant protein?)”

Response: This is indeed an important consideration. Although Cys48Ser mice are generally healthy, their extracted LS does not show normal surface activity, and thus there is likely some role for covalent dimerization even if it’s not essential for life. To better point to this nuance, we’ve added a reference that explores the role of homodimers and modified this part of Section 2.1 to read, “LS from mice with the Cys48Ser mutation shows altered LS activity although the mice remain healthy perhaps, as has been suggested because SP-B dimerization can also occur via non-covalent interactions.”

Point5: “When carrying out far-UV CD, it’s not fully clear how the protein concentration was determined, nor why the particular methodology chosen was used to infer helical content. Given that there are, for example, deconvolution datasets specific to membrane proteins (recognizing that SP-B isn’t an integral membrane protein), how different might those results be? How does the percentage of helicity inferred from evaluating normalized CD spectra compare to the expectations based on secondary structure predictions shown in the schematic figures? And could the dramatic differences in helical percentage in different environments be an artifact of the helical estimation methodology used rather than a true representation of differences in helicity in these environments? If the disulfide bonds are indeed formed (point 3) it seems hard to rationalize such extreme changes in helical content.”

Response: We were remiss in not mentioning that the protein concentrations were cross-checked between fluorescamine, BCA, Biuret, Absorbance at 280 and Bradford. This information has been added to the methods.

Given rSP-B’s positive charge of +7 per chain it is not surprising to us that its helicity is very different between DPC and DPC/SDS. Such findings also parallel common observations for positively charged antimicrobial peptides that are often most helical in bilayers with anionic lipids. We are a little more puzzled by the intermediate helicity observed for LMPG, but 100% anionic lipid is a very unlike lung surfactant which has only 10% anionic lipid [5] – we thus think it’s better to remove the 100% LMPG data set.

With regards estimation of helical content in rSP-B we present below a deconvolution analysis. Since rSP-B is a bit of a one-of a kind protein there may not be any perfect basis sets for it, but the results blow indicate similar % helicity from deconvolution compared to our simple Rohl and Baldwin estimate – though we may be underestimating the helicity by ~5%. However, the ratio of helicity in DPC/SDS compared to DPC is very similar for all the methods (3.2 to 3.6 times higher)

Condition Method Helix 1 (%) Helix 2 (d*) (%) Strand 1 (%) Strand (d*) 2 (%) Turn (%) Unordered (%) Total Helicity (%) Rohl & Baldwin Estimate (%) NRMSD

0.2% DPC/SDS (9:1) Selcon 3 (Set 4) 33.1 20.2 0.0 0.0 15.4 26.3 53.3 0.248 (Valid)

Contin-LL (SMP180t) 35.7 18.9 0.0 0.0 10.4 31.2 54.6 47% 0.084 (Valid)

K2D 59.0 -- 7.0 -- -- 33.0 59.0 0.231 (Invalid)

0.2% LMPG Selcon3 (SMP180t) 19.5 13.4 13.3 7.1 10.3 32.0 32.5 0.243 (Valid)

Contin-LL (SMP180t) 21.6 9.8 17.9 8.9 9.5 32.4 32.2 25% 0.157 (Valid)

K2D 28.0 -- 37.0 -- -- 35.0 28.0 0.122 (Valid)

0.5% DPC Contin-LL (SMP180t) 7.4 7.9 22.5 11.6 11.5 39.1 15.3 0.074 (Valid)

Selcon 3 (SMP180t) N/A N/A N/A N/A N/A N/A N/A 13% No output

K2D 18.0 -- 30.0 -- -- 52.0 18.0 0.578 (Invalid, >0.227)

d* = disordered

Point6: “Based on the speculation that bacterial proteases are leading to products of the approximate MW of the SP-B, are there any insights from the sequence of the fusion protein as to which protease(s) might be doing so and what the cleavage product would be?”

Response: While we have not performed in-silico analysis to identify protease recognition sites in the fusion protein sequence, it is possible that proteases in the E. coli system could cleave non-specifically at exposed or flexible regions of the fusion protein, especially near the junction between SN and rSP-B. Common E. coli proteases, such as Lon, ClpXP, OmpT, and DegP, which are known to cleave non-specifically at exposed sites in recombinant proteins including aggregated proteins and proteins in inclusion bodies could act at these sites. References 64 and 65 have been added to this sentence to back this up.

Point7: “The figure caption for Fig. 4 (B) is a little unclear (i.e., what is the “different” gel that is referred to? How was the monoclonal antibody detected?)”.

Response: The term "different" gel in Fig. 4B refers to the duplicate gel used for Western blotting, which was run separately from the gel shown in panel A. The monoclonal anti-SP-B antibody was detected using an anti-mouse secondary antibody, followed by colorimetric detection. We have updated the figure caption for clarity.

Point8: “ It’s unclear what’s meant by “only the near 8.5 kDa band material binds to the monoclonal anti-SP-B mouse antibody”.

Response: Thank you for your comment. The phrase "only the near 8.5 kDa band material binds to the monoclonal anti-SP-B mouse antibody" refers to the fact that, after CNBr cleavage of the fusion protein, a band of approximately 8.5 kDa (which corresponds to the rSP-B fragment) is specifically recognized by the anti-SP-B monoclonal antibody in the Western blot. This confirms that the expected rSP-B cleavage product, and not other non-specific bands, is binding to the antibody. We have revised the manuscript by removing the word “near” to clarify that the 8.5 kDa band is indeed the one detected by the antibody.

Point 9: “The authors consistently refer to the protein as showing “some” surface activity. This seems an odd choice of wording and perhaps could be either made more specific in nature or just referred to as being surface active?”

Response: We agree that the term “some” surface activity may seem vague. To improve clarity, we have revised the manuscript to describe the protein as being “surface active” rather than using the term “some surface activity.”

Minor Comments:

Point 10: “It would be good to provide reference(s) for the statement that the anomalous migration by SDS-PAGE is expected (p. 9)”

Response: Good point. We’ve added two references and a new sentence to further explain and back up this point: “Hydrophobic proteins, especially membrane-associated ones, often exhibit anomalous migration in SDS-PAGE due to their interactions with detergents and the presence of disulfide bonds, which can alter migration patterns [62, 63].”

Point 11: “Abstract and again on p. 7 – italicize Staphylococcus?”

Response: Thank you for pointing this out. We have corrected the manuscript by italicizing Staphylococcus in the abstract and on page 7.

Point 12: “Abstract – specify that C43 is an Escherichia coli strain. (See also comment about whether it is suitable to generalize statement finding of intracellular membrane production.)”

Response: We have revised the abstract to specify that Escherichia coli strain C43 cells was used for the expression of rSP-B (Δ7NTC48S-SP-B-6His). Regarding the generalization of our findings to intracellular conditions, we have now addressed this concern in response to Point 1.

Point 13: “p. 7, para. 1 – correct “presented in this work presents”; spell out Escherichia on first use [here]”

Response: We have corrected the redundancy in the sentence and revised it to: "The SP-B protein presented in this work is, unlike the one in [51] ...”. Additionally, we have used "Escherichia coli" on its first mention in the paragraph as suggested.

Point 14: “Fig. 2 caption – capitalize “Amino” at start of (B)”

Response: Fixed

Point 15: “p. 13, first para. of discussion – probably more suitable to refer to this as “first report” rather than “first publication”?; second sentence – remove comma after permit?”

Response: Done

Point 16: “p. 14, 2nd para. – not fully clear how an MD simulation (presumably MD, rather than “molecular simulation”?) “might” show cholesterol binding sites; the entire sentence is worded rather speculatively.”

Response: We have revised the sentence. The updated version reflects the potential findings from the molecular dynamics study, without overstating the conclusions.

Point 17: “Methods are fairly well described, but some additional references for sources and precedents for key protocols would be beneficial to include”

Response: We have reviewed the methods section and added relevant references to key protocols where applicable. These references provide additional context and precedents for the experimental procedures described in the manuscript. We believe these additions will enhance the clarity and reproducibility of the methods used.”

Point 18: “Section 5.7 – include “spectroscopy” in title? Rate of data collection and pitch? Was buffer/blank correction employed? (“Baseline correction” is mentioned, but this has a different meaning and it’s not fully clear how this would be carried out in CD?):

Response: To clarify the methods used in Section 5.7, we have revised the description to include additional information on the data collection rate and pitch, as well as explanation regarding the baseline correction. The term "baseline correction" refers to the subtraction of a buffer/blank spectrum as described in Greenfield et al., 2006 [85] and this procedure was carried out by subtracting the signal of the buffer or solvent from the sample spectrum to remove contributions unrelated to the peptide.

Point 19: “Section 5.8 – Specify measurement being carried out in the section title rather than instrument; the wt% per mg/mL of lipids in a v/v solution is very hard to follow - perhaps this could be described in a different way?; I believe that units for water should be MOhm x cm (not just MOhm)?; It would also be good to clarify what was done with the three measurements that were carried out for each sample (i.e., are these cyclic responses that are considered individually or were these averaged, or some other treatment?)”

Response: We have revised Section 5.8 to clarify these points and also fixed the water quality units.

Reviewer 2

“The manuscript describes to my knowledge the first recombinant production and purification of (almost) full-length human SP-B. Results are valuable for basic research on this protein and potentially for other, including therapeutic, applications.”

Point 1: “ Expression (p17): Culture volume? Type of culture (e.g. shake flask)? Wet cell mass would also be helpful to know.”

Response: We’ve added the culture type and volume to section 5.2. We have had a look through our lab notebooks and can’t find that we’ve recorded wet cell mass

---

## [Decision Letter · Decision Letter 1]

7 Mar 2025

Bacterial expression, purification and folding of exceptionally hydrophobic and essential protein: Surfactant Protein-B (SP-B)

PONE-D-24-45785R1

Dear Dr. Booth,

We’re pleased to inform you that your manuscript has been judged scientifically suitable for publication and will be formally accepted for publication once it meets all outstanding technical requirements.

Kind regards,

Michael Massiah

Academic Editor

PLOS ONE

Additional Editor Comments (optional):

Reviewers' comments:

Reviewer's Responses to Questions

**Comments to the Author**

1. If the authors have adequately addressed your comments raised in a previous round of review and you feel that this manuscript is now acceptable for publication, you may indicate that here to bypass the “Comments to the Author” section, enter your conflict of interest statement in the “Confidential to Editor” section, and submit your "Accept" recommendation.

Reviewer #1: All comments have been addressed

Reviewer #2: All comments have been addressed

2. Is the manuscript technically sound, and do the data support the conclusions?

Reviewer #1: Yes

Reviewer #2: (No Response)

3. Has the statistical analysis been performed appropriately and rigorously? 

Reviewer #1: N/A

Reviewer #2: (No Response)

4. Have the authors made all data underlying the findings in their manuscript fully available?

Reviewer #1: Yes

Reviewer #2: (No Response)

5. Is the manuscript presented in an intelligible fashion and written in standard English?

Reviewer #1: Yes

Reviewer #2: (No Response)

6. Review Comments to the Author

Reviewer #1: All concerns and comments following from the initial submission have been comprehensively and very conscientiously addressed by the authors. I feel this manuscript is now suitable for publication.

Reviewer #2: (No Response)

7. PLOS authors have the option to publish the peer review history of their article (what does this mean? ). If published, this will include your full peer review and any attached files.

**Do you want your identity to be public for this peer review?** For information about this choice, including consent withdrawal, please see our Privacy Policy .

Reviewer #1: No

Reviewer #2: No

---

## [Editor Report · Acceptance letter]

PONE-D-24-45785R1

PLOS ONE

Dear Dr. Booth,

I'm pleased to inform you that your manuscript has been deemed suitable for publication in PLOS ONE. Congratulations! Your manuscript is now being handed over to our production team.

Kind regards,

on behalf of

Dr. Michael Massiah

Academic Editor

PLOS ONE
